# Extended-spectrum beta-lactamase-producing *Escherichia coli* and *Klebsiella pneumoniae* from human carriage, the human-polluted environment, and food: Molecular epidemiology of two prospective cohorts in five European metropolitan areas

Tess D. Verschuuren[1,2] *, Julia Guther[3], Maria Eugenia Riccio[4], Daniel Martak[5,6], Elena Salamanca[7,8,9], Siri Göpel[10], Nadine Conzelmann[10], Jelle Scharringa[11], Patrick Musicha[12], Ingo B. Autenrieth[13], Ben S. Cooper[12], Didier Hocquet[5,6], Evelina Tacconelli[8,14], Jesús Rodríguez-Baño[7,8,9], Stephan Harbarth[4], Ad C. Fluit[11], Silke Peter[3], Jan A. J. W. Kluytmans[2,11]

1 Mahidol-Oxford Tropical Medicine Research Unit, Faculty of Tropical Medicine, Mahidol University, Bangkok, Thailand, 2 Julius Centre for Health Sciences and Primary Care, University Medical Centre Utrecht, Utrecht, The Netherlands, 3 Institute of Medical Microbiology and Hygiene, University Hospital Tübingen, Tübingen, Germany, 4 Infection Control Programme and Division of Infectious Diseases, University of Geneva Hospitals and Faculty of Medicine, Geneva, Switzerland, 5 Infection Control Unit, Centre de Ressources Biologiques - Filière Microbiologique de Besançon, Centre Hospitalier Universitaire, Besançon, France, 6 Chrono-Environnement UMR 6249, CNRS, Université Bourgogne Franche-Comté, Besançon, France, 7 Department of Infectious Diseases and Clinical Microbiology, University Hospital Virgen Macarena, Seville, Spain, 8 Department of Medicine, University of Sevilla/ Biomedicines Institute of Sevilla (IBiS), Sevilla, Spain, 9 Centro de Investigación Biomédica en Red en Enfermedades Infecciosas (CIBERINFEC), Madrid, Spain, 10 Infectious Diseases, Department of Internal Medicine, University Hospital Tübingen, Tübingen, Germany, 11 Department of Medical Microbiology, University Medical Centre Utrecht, Utrecht, The Netherlands, 12 Centre for Tropical Medicine & Global Health, Nuffield Department of Medicine, University of Oxford, Oxford, United Kingdom, 13 University Hospital Heidelberg, Heidelberg, Germany, 14 Infectious Diseases, Department of Diagnostics and Public Health, University of Verona, Verona, Italy

* t.d.verschuuren@gmail.com

## Abstract

### Objectives

For 475 ESBL-producing *Escherichia coli* (ESBL-Ec), and 171 ESBL-producing *Klebsiella pneumoniae* (ESBL-Kp) collected from human carriers, the human-polluted (hp)-environment, and food: (i) to compare the antimicrobial resistance gene (ARG) content, and (ii) to assess clonal relationships between human and non-human isolates.

### Materials and methods

Two prospective multicenter cohorts were assessed: colonized hospitalized index-subjects and household contacts, and long-term care facility (LTCF) residents.

**Data availability statement:** Raw reads are available on the European Nucleotide Archive (ENA), under the accession number PRJEB50545. Metadata with accession numbers are available in the supplementary material. Online access interactive maximum-likelihood trees: ESBL-Ec: https://microreact.org/project/wSXSAfjgAH1Xr9af46Hbcb-modern-project-esbl-e-coli-n475, ESBL-Kp: https://microreact.org/project/vRJHCdos8zW-cv4yHMTKjak-modern-project-esbl-k-pneumoniae.

**Funding:** "The MODERN studies (Understanding and modelling reservoirs, vehicles and transmission of ESBL-producing Enterobacteriaceae in the community and long term care facilities) were funded under the Joint Programming Initiative on Antimicrobial Resistance 2016 Joint Call. Funding was provided by the Netherlands Organization for Health Research and Development (ZonMw) (grant nos. 681055 and 547001004 to T.D.V., J.S., A.C.F., J.A.J.W.K.); the Swiss National Science Foundation (grant no. 40AR40-173608 to M.E.R., S.H.); the German Federal Ministry of Education and Research (grant no. 01KI1701 to J.G., S.G., N.C., I.B.A., E.T., S.P.); the French Agence Nationale de la Recherche (grant no. ANR-16-JPEC-0007-03 to D.M., D.H.); the UK Medical Research Council (grant no. MR/R004536/1 to P.M., B.S.C.); and the Instituto de Salud Carlos III (grant no. AC16/00076 to E.S., J.R.B.). E.S. and J.R.B. also received support from the Spanish Network for Research in Infectious Diseases (REIPI RD16/0016/0001), co-financed by the European Development Regional Fund. There was no additional external funding received for this study.".

Additionally, linked hp-environment and food samples were collected. Presence of ARGs were assessed using pairwise comparisons and proportional similarity index (PSI). Clonal relationships were assessed using cgMLST distance visualizations and maximum likelihood phylogeny.

## Results

ESBL-Ec and ESBL-Kp co-occurred in 14/65 households, 3/6 LTCFs, and in 33/202 of ESBL-positive participants. Thirty-nine percent of detected ARG types were found in both species (36/93). Frequencies of beta-lactamase, ESBL, aminoglycoside, and sulfonamide ARG types from human ESBL-Ec and ESBL-Kp overlapped considerably: PSIs 0.59–0.75, and were equal or higher compared to the overlap between ESBL-Ec from humans and food isolates: PSIs 0.33–0.72. Isolates from humans and the hp-environment were frequently clonally related, indicating human contamination of the environment. Links with food isolates were observed less frequently. For ESBL-Ec both interregional and regional clonal dissemination were observed, while for ESBL-Kp clonal dissemination was mainly regional.

## Conclusions

ESBL-Ec and ESBL-Kp from human carriage showed considerable overlap in ARG content. Furthermore, clonal links were observed frequently between humans and hp-environment, and with lower frequency between humans and food. These findings are consistent with human-to-human transmission as an important driver of ARG spread in humans.

## Introduction

Antimicrobial resistance was declared as one of the top 10 public health threats facing humanity by the World Health Organization in 2019 [1]. In Europe, extended-spectrum beta-lactamase-producing *Escherichia coli* (ESBL-Ec) and *Klebsiella pneumoniae* (ESBL-Kp) are of particular concern, as these species are the most frequent causative agents of infections with antimicrobial resistant pathogens [2]. Furthermore, previous studies have shown that intestinal carriage of ESBL-producing Enterobacterales will result in infection in 8% of carriers [3]. Scientific effort has been put into elucidating reservoirs of human carriage for these bacteria, but the data on the relevance of sources outside humans are conflicting [4–7]. Most antimicrobial resistance genes (ARGs) that have been observed in *E. coli* also occur in *K. pneumoniae*, and vice versa, which is indicative of horizontal gene transfer (HGT) [4,6,8,9]. However, ARG types have rarely been compared systematically between multidrug-resistant *E. coli* and *K. pneumoniae* [10–14]. Five studies have been performed in a retrospective way and were mostly single-center (*n* = 3), with relatively small sample sizes (*n* = 71–150) [10–14]. Here, we statistically compare the ARG types occurring in ESBL-Ec and ESBL-Kp from human carriers with a close relationship to healthcare,

the human-polluted (hp)-environment, and food. Additionally, we assess clonal relationships between human and non-human isolates.

## Materials and methods

### Study design, and data-collection

Two prospective cohort studies were conducted between November 1st 2017 and August 31st 2019 in five European metropolitan areas: Besançon (France), Geneva (Switzerland), Seville (Spain), Tübingen (Germany), and Utrecht (the Netherlands). Data collection, sample processing, and microbiological methods have been described previously [7,15,16]. Briefly, the household study recruited ESBL-Ec- and ESBL-Kp-positive index patients during hospitalization. Participants were followed up at home for four months after discharge, together with ≥1 other household member, and provided a fecal sample at four time points during follow-up. In the long-term care facility (LTCF) study, participating residents were followed for eight months, and provided a fecal sample or perianal swab at eight time points during follow-up (S1, p1-2). Additionally, food and hp-environment samples were collected: (i) food was sampled at three time points in supermarkets where participating households reported shopping, and eight times in the kitchen of the participating LTCFs, (ii) samples from the hp-environment were collected from: a) LTCF U-bends, b) LTCF surfaces, c) LTCF wastewater, d) wastewater treatment plant (WWTP) inflow connected to the LTCF, and e) river downstream to the WWTP. Reservoirs a) and b) were sampled twice, the other reservoirs eight times (S1, p1-2).

### Ethics

The institutional review board of the university hospitals of Besançon, Geneva, Seville, Tübingen, and Utrecht reviewed, and approved the studies or waived the need for further ethical review. In Geneva only the household cohort study was conducted, and thus reviewed. All enrolled participants or their legal representatives provided written informed consent for participation in this study.

### Microbiological methods

The microbiology laboratory of each participating center used selective culture media, and selective enrichment broth with standardized methods for the collected human, environmental surface, wastewater, and food samples. Subsequently, identification of ESBL-Ec and ESBL-Kp was performed (S1, p3).

### Sequencing and selection of unique isolates

Isolates identified as ESBL-Ec or ESBL-Kp were shipped to Tübingen or Utrecht. DNA isolation (DNeasy UltraClean Microbial Kit, Qiagen), sequencing (NextSeq and MiSeq platforms, Illumina, San Diego, USA), and *de novo* assembly (Spades v3.11.1) was performed on all isolates. Quality of assembled sequences was assessed (S1, p3). In this study, only unique isolates were included (S1, p3).

### *In silico* molecular typing

ARGs conferring resistance to clinically relevant antimicrobials were identified using ResFinder (v3.2) (S1, p3) [17]. These included ARGs conferring resistance to: (i) penicillins only (small- or broad-spectrum), from here on referred to as beta-lactams, (ii) penicillins and cephalosporins, from here referred to as ESBLs, (iii) penicillins and carbapenems, referred to as carbapenemases, (iv) fluoroquinolones, (v) aminoglycosides, (vi) fosfomycin, (vii) trimethoprim, (viii) sulfonamides, and lastly (ix) colistin. Classifications were made according to genotype-phenotype (antimicrobial and class) translations available in the ResFinder database (https://bitbucket.org/genomicepidemiology/resfinder_db/src/master/phenotypes.txt) and EUCAST Clinical Breakpoint Tables (v13.0) to further subgroup the 'β-lactam' class provided by

ResFinder [18,19]. A list of all detected ARG types with classifications is included in the supplement (S1, p4). Sequence types (STs) were identified using published tools, and sub-clades were assigned to ST131 (ESBL-Ec) isolates (S1, p3) [20,21].

## Analysis

The proportions of the 10 most frequently occurring ARG types per class were compared using a two-proportion z-test. The pairwise overlap between sampled reservoirs of acquired ARG type distributions was quantified with Czekanowski's proportional similarity index [4].

$$PSI = 1 - 0.5 \sum k \left| p(reservoir[n])k - q(reservoir[nx])k \right|$$

Where $p$ corresponded to the relative frequency of gene type $k$ in reservoir $n$, and $q$ corresponded to the relative frequency of the same gene type in reservoir $nx$ [4,22]. The denominator of the relative frequency was the total number of genes in the corresponding ARG class. The PSI is a proportion, with 0 being interpreted as no overlap, and 1 as perfect overlap in ARG type distributions between two reservoirs. Bootstrap iterations (5,000) were used to calculate 95% confidence intervals (CIs) (boot R-package, v1.3.23) [4]. Clonal transmission between humans was described previously [15]. Here, we assessed clonal relationships between human and non-human isolates, with a visualization where isolates were ordered based on epidemiological setting (qgraph R-package, v1.9.4). To prevent detection of spurious relationships between human and non-human isolates, a cgMLST threshold was chosen for ESBL-Ec of ≤0.0040 (~10 alleles) for definition of clonally related pairs, based on previous studies [7,22]. The threshold for ESBL-Kp was set at ≤0.0035 (~10 alleles) [22,23]. A sensitivity analysis with increased thresholds was performed. Lastly, maximum likelihood (ML) trees were created for ESBL-Ec and ESBL-Kp (S1, p5).

## Results

In total, 196 participants were included (110 household members, and 86 LTCF residents), carrying 205 ESBL-Ec, and 101 ESBL-Kp isolates (Table 1). Furthermore, 232 hp-environment and 108 food isolates were included, consisting of 270 ESBL-Ec, and 70 ESBL-Kp isolates (Table 1). A list of all included strains is provided in the supplement (S2). During follow-up, both species were detected in human samples from 14/65 (22%) households, and 3/6 LTCFs, and 17% of participants were colonized with both ESBL-Ec and ESBL-Kp. A mean of 1.6 isolates (range: 1–10) were detected in participants during follow-up (Table 1). On average, participants from Utrecht were colonized with 1.1 unique isolates, while participants from Seville were colonized with 1.7 unique isolates (p<0.0001), mostly due to more frequent carriage of ESBL-Kp isolates.

### Comparison of ARG content

Of a total of 93 detected ARG types, 39% ($n = 36$) were detected in both species, demonstrating overlap of resistance genes from all classes, except carbapenems and colistin (Table 2). ARGs encoding for carbapenemases (ESBL-Ec ($n = 2$): $bla_{OXA-48}$, $bla_{OXA-181}$, ESBL-Kp: $bla_{KPC-2}$ ($n = 2$)) and colistin-resistance (ESBL-Kp: $mcr$-$1$ ($n = 1$)) were rare. When assessing the 10 most frequently occurring ARG types per class, we observed similar frequencies for ESBL-Ec and ESBL-Kp from human isolates. However, overall proportions of ARGs were higher for ESBL-Kp (Fig 1 and S1, p7). For example, compared to ESBL-Ec, ESBL-Kp were more likely to harbor ARGs from the $bla_{SHV}$ family (77% vs 12%), resistance genes $aac(6')$-$Ib$-$cr$ (51% vs 14%), $aac(3)$-$IIa$ (45% vs 8%), and $bla_{OXA-1}$ (50% vs 15%), and $fosA$ genes (75% vs 1%) (Fig 1 and S1, p7). When assessing the sampled reservoirs separately, ESBL-Ec from food showed differences in detected ARG types compared to other groups. For example, ESBL-Ec from food harbored less $bla_{OXA-1}$ (2% vs 28%), $bla_{CTX-M-15}$ (12% vs 59%), $aac(6')$-$Ib$-$cr$ (2% vs 27%), $aac(3)$-$IIa$ (2% vs 21%), and more $bla_{SHV-12}$ (39% vs 6%), $bla_{CTX-M-1}$

**Table 1. Characteristics of the included first unique ESBL-producing *E. coli* and *K. pneumoniae* isolates from humans, the human-polluted environment, and food.**

| | ESBL+ | ESBL-Ec | ESBL-Kp |
|---|---|---|---|
| ESBL-positive participants[a] (n) | 196 | 151 | 78 |
| household members | 110 | 82 | 41 |
| LTCF-residents | 86 | 69 | 37 |
| Number of unique isolates per participant (mean; range) | 1.6 (1-10) | 1.1 (0-7) | 0.52 (0-3) |
| household members[b] | 1.4 (1-10) | 1.0 (0-7) | 0.43 (0-3) |
| LTCF residents[c] | 1.7 (1-5) | 1.1 (0-5) | 0.60 (0-3) |
| Unique isolates (n) | 646 | 475 | 171 |
| human | 306 | 205 | 101 |
| human-polluted environment | 232 | 175 | 57 |
| food | 108 | 95 | 13 |
| Composition human-polluted environment (n) | | | |
| LTCF surface | 15 | 7 | 8 |
| LTCF U-bend | 26 | 13 | 13 |
| LTCF wastewater outflow | 34 | 19 | 15[d] |
| wastewater treatment plant inflow | 73 | 64 | 9 |
| downstream river | 84 | 72 | 12 |

[a]n: number of participants carrying ESBL-*E. coli* or ESBL-*K. pneumoniae* on at least one time-point. [b]The maximum follow-up of a household member was four months. [c]The maximum follow-up of LTCF residents was eight months. [d]seven isolates were retrospectively identified as *Klebsiella variicola*.

(32% vs 5%), *ant(3")-Ia* (38% vs 9%), and *sul3* (26% vs 3%) compared to the cumulative frequencies in other groups (Fig 1 and S1, p7).

The PSI was calculated in order to quantify the total overlap of observed ARG types (grouped per antimicrobial class) between the sampled reservoirs. Overall, the ARG types from the same species observed in human and human-polluted environment isolates were most similar, ranging between 0.76 (95% CI 0.7–0.8) for beta-lactams from ESBL-Kp to 0.94 (0.9–1.0) for fosfomycin from ESBL-Ec (Fig 2 and S1, p8). However, ESBL-Ec and ESBL-Kp from humans also showed considerable similarity in beta-lactams (PSI 0.63; 0.5–0.7), ESBLs (PSI 0.59; 0.5–0.7), aminoglycosides (PSI 0.68; 0.6–0.7), and sulfonamides (PSI 0.75; 0.6–0.9) ARG types. In fact, for these ARGs, PSIs were equal or higher to PSIs from humans and food harboring ESBL-Ec. Lastly, less overlap was observed between ESBL-Ec and ESBL-Kp from humans for fluoroquinolone (PSI 0.37; 0.3–0.5), fosfomycin (PSI 0.24; 0.1–0.3), and trimethoprim (PSI 0.36; 0.3–0.5) ARG types (Fig 2 and S1, p8).

## Comparison of core genome content of isolates from humans, the hp-environment, and food

The three most frequent STs within ESBL-Ec were: ST131 (28%; 44% in human isolates, 23% hp-environment, 0% food), ST10 (11%; 10% human, 10% hp-environment, 12% food), ST69 (5.1%; 2.8% human, 5.1% hp-environment, 9.5% food). For ESBL-Kp, the three most frequent STs were ST405 (22%; 27% in human isolates, 18% hp-environment, 0% food), ST307 (10%; 10% human, 12% hp-environment, 0% food), ST323 (5.8%; 3.0% human, 12% hp-environment, 0% food). During follow-up, eight participants were colonized with >1 ST131 isolate, often harboring different ESBL genes (S1, p6).

A pattern of clonally related isolates between humans and the hp-environment was observed (Fig 3a-b). This observation was most pronounced in Seville for both ESBL-Ec and ESBL-Kp, where isolates from residents were regularly related to LTCF surfaces, U-bends, and LTCF wastewater outflow (Fig 3c-d). These connections reflected spread from LTCF residents to the LTCF-associated environments, and potentially human acquisition through contaminated surfaces (and U-bends). Furthermore, similar patterns were observed for ESBL-Ec isolates from Besançon, Tübingen, and Geneva (Fig 3a, and

**Table 2. Detected antimicrobial resistance gene (ARG) types grouped per antimicrobial class of conferred resistance.**

| | ESBL-Ec | ESBL-Kp | Both species |
|---|---|---|---|
| Antimicrobial class | ARG type | ARG type | ARG type |
| beta-lactams[a] | blaCARB-2, blaDHA-1, blaOXA-1, blaOXA-2, blaTEM-1, blaTEM-122, blaTEM-2, blaTEM210, blaTEM-33 | blaDHA-1, blaLEN-24, blaOKP-A-12, blaOKPA3, blaOKP-B-1, blaOKP-B-15, blaOKP-B-3, blaOKP-B-9, blaOXA-1, blaSCO-1, blaSHV-1, blaSHV-11, blaSHV-110, blaSHV145, blaSHV-172, blaSHV-178, blaSHV-187, blaSHV-27, blaSHV-62, blaSHV75, blaSHV-76, blaTEM-1 | blaDHA-1, blaOXA-1, blaTEM-1 |
| ESBLs | blaCTX-M-1, blaCTX-M-14, blaCTX-M-15, blaCTX-M-27, blaCTX-M-3, blaCTX-M-32, blaCTX-M-55, blaCTX-M-65, blaCTX-M-8, blaSHV-12, blaSHV-2, blaTEM-106, blaTEM-52 | blaCTX-M-1, blaCTX-M-14, blaCTX-M-15, blaCTX-M-209, blaCTX-M-27, blaCTX-M-3, blaCTX-M-9, blaSHV-106, blaSHV-12, blaSHV129, blaSHV-13, blaSHV-2, blaSHV-5, blaTEM-169 | blaCTX-M-1, blaCTX-M-14, blaCTX-M-15, blaCTX-M-27, blaCTX-M-3, blaSHV-12, blaSHV-2 |
| carbapenems | blaOXA-181, blaOXA-48 | blaKPC-2 | – |
| fluoroquinolones | aac(6')-Ib-cr, qepA4, qnrB1, qnrB19, qnrB4, qnrS1, qnrS2 | aac(6')-Ib-cr, qnrA1, qnrB1, qnrB19, qnrB4, qnrB52, qnrS1, qnrS2 | aac(6')-Ib-cr, qnrB1, qnrB19, qnrB4, qnrS1, qnrS2 |
| aminoglycosides | aac(3)-IIa, aac(3)-IId, aac(3)-IVa, aac(3)-VIa, aac(6')-Ib-cr, aadA1, aadA2, aadA5, ant(2'')-Ia, ant(3'')Ia, aph(3')-Ia, aph(3'')-Ib, aph(3')-Id, aph(4)-Ia, aph(6)-Id | aac(3)-IIa, aac(3)-IId, aac(6')-Ib-cr, aadA1, aadA16, aadA2, aadA4, aadA5, ant(2'')-Ia, ant(3'')-Ia, aph(3')-Ia, aph(3'')-Ib, aph(6)-Id | aac(3)-IIa, aac(3)-Iid, aac(6')-Ib-cr, aadA1, aadA2, aadA5, ant(2'')-Ia, ant(3'')-Ia, aph(3')-Ia, aph(3'')-Ib, aph(6)-Id |
| fosfomycin | fosA, fosA3 | fosA, fosA6, fosA5 | fosA |
| trimethoprim | dfrA1, dfrA12, dfrA14, dfrA16, dfrA17, dfrA5, dfrA7, dfrA8 | dfrA1, dfrA12, dfrA14, dfrA15, dfrA16, dfrA17, dfrA27 | dfrA1, dfrA12, dfrA14, dfrA16, dfrA17 |
| sulfonamides | sul1, sul2, sul3 | sul1, sul2, sul3 | sul1, sul2, sul3 |
| colistin | – | mcr-1 | – |

[a]Includes antimicrobial resistance genes conferring resistance to small or broad-spectrum penicillins, excluding ESBLs and carbapenemases.

S1, p9). Observed clonally related isolates from humans and WWTP inflow or downstream rivers likely reflected spread of certain genetically conserved strains in a larger geographical area. Lastly, we found two clonally related human-food pairs for ESBL-Ec, both without a known epidemiological link (Fig 3a).

The ML-tree of ESBL-Ec showed no evident clustering based on sample group, with the exception of the absence of food isolates within the ST131 and ST1193 phylogeny (Fig 4a). Both STs were retrieved in all cities, potentially indicating clonal human-to-human spread in a large geographic area. For ST131, clade A (24/138), B (5/138), and C (108/138) clustered separately, and most of these isolates carried the $bla_{CTX-M-15}$, $bla_{CTX-M-27}$, or $bla_{CTX-M-14}$ $bla_{ESBL}$ (Fig 4a). Clade C1 and C2 were distributed evenly with 51 and 57 isolates, respectively. Most of C1 isolates carried $bla_{CTX-M-27}$ ($n=40$). For ESBL-Kp, most STs were specific to a metropolitan area, likely reflecting local clonal dissemination (Fig 4b). However, ST405 and ST323 were observed in two countries, and ST219 ($n=6$) was observed in four countries from human, river, and chicken and turkey isolates, potentially reflecting widespread clonal dissemination. The few food-related isolates ($n=13$) were scattered throughout the tree (Fig 4b).

ESBL-Kp: https://microreact.org/project/vRJHCdos8zWcv4yHMTKjak-modern-project-esbl-k-pneumoniae.

## Discussion

In this multicentre prospective study, ESBL-Ec and ESBL-Kp regularly co-occurred in humans who, by the design of this study, were in close contact with healthcare. The majority of $bla_{ESBL}$ types were found in both species. $bla_{ESBL}$ distributions of ESBL-Ec and ESBL-Kp sampled from humans were more similar than ESBL-Ec sampled from humans and food. Non-ESBL-ARGs occurrence was highest in ESBL-Kp from any source and lowest in ESBL-Ec from food. While ESBL-Kp did

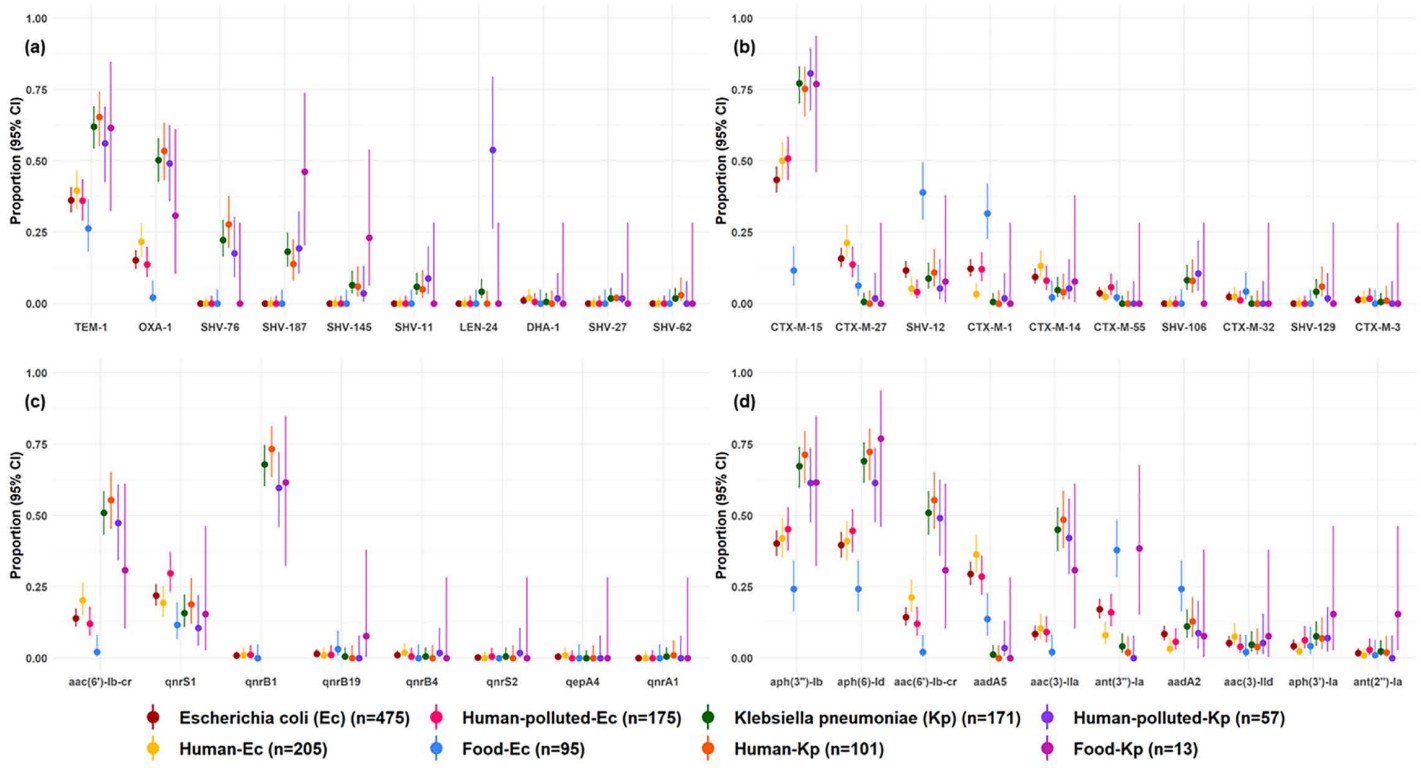

**Fig 1. Proportions (95% CI) of the ten most frequent antimicrobial resistance gene types per isolate.** Panel (a) beta-lactamases, panel **(b)** ESBLs, panel (c) fluoroquinolones, panel (d) aminoglycosides.

more often carry ARGs, substantial overlap was observed for ARGs encoding resistance to beta-lactams, and aminoglycosides between ESBL-Ec and ESBL-Kp from humans.

A relatively low similarity in $bla_{ESBL}$ type distributions was observed between ESBL-Ec from food and humans, which was mainly explained by a lower occurrence of $bla_{CTX-M-15}$, and higher occurrence of $bla_{CTX-M-1}$ and $bla_{SHV-12}$ in food [4]. Furthermore, ST131, the most prevalent ST in the dataset, was absent in food, and ARG proportions were lowest in food-related ESBL-Ec. Early research showed potential links between ESBL-Ec from food and humans [5]. However, more recent studies found that, at least in high income countries, ESBL-Ec and ESBL-Kp mainly spread through human-to-human transmission, while transmission of ESBL-Ec from food is less frequently observed and does probably not play an important role for ST131 the most frequently observed ST in humans. The role of food can be considered as spill-over events (potentially introducing new resistance traits into humans which may subsequently spread) [4,6,23].

The $bla_{CTX-M-15}$ gene has been widely observed as the most frequently occurring $bla_{ESBL}$ in both *E. coli* and *K. pneumoniae* from human carriage and infection [4,6,20]. In this study, the majority of the detected $bla_{ESBL}$ types occurred in both species, with $bla_{CTX-M-15}$ detected in more than half of the included isolates. Furthermore, we observed considerable similarity in ARG content encoding resistance to beta-lactams, aminoglycosides, and sulfonamides. These observations are supported by studies that statistically compared ARG content between *E. coli* and *K. pneumoniae*, demonstrating overlap in ARGs conferring resistance to several antimicrobial classes [10–14].

Hospital and LTCF environments are well-established as significant reservoirs and amplifiers of multi-drug resistant bacteria, contributing to the persistence and spread of these organisms within healthcare settings and the broader community [16,24,25]. To our knowledge this is the first study to genetically compare ESBL-Ec and ESBL-Kp that were

| (a) | EC H | EC HP | EC F | KP H | KP HP | KP F |
|---|---|---|---|---|---|---|
| EC H | - | 0.82 .7-.9 | 0.63 .5-.7 | 0.63 .5-.7 | 0.56 .5-.7 | 0.51 .3-.7 |
| EC HP | | - | 0.78 .7-.9 | 0.53 .4-.6 | 0.47 .4-.6 | 0.45 .3-.6 |
| EC F | | | - | 0.37 .3-.5 | 0.32 .2-.4 | 0.30 .1-.5 |
| KP H | | | | - | 0.76 .7-.8 | 0.58 .4-.7 |
| KP HP | | | | | - | 0.57 .4-.7 |

| (b) | EC H | EC HP | EC F | KP H | KP HP | KP F |
|---|---|---|---|---|---|---|
| EC H | - | 0.80 .7-.9 | 0.33 .2-.4 | 0.59 .5-.7 | 0.61 .5-.7 | 0.58 .3-.8 |
| EC HP | | - | 0.40 .3-.5 | 0.59 .5-.7 | 0.61 .5-.7 | 0.56 .3-.7 |
| EC F | | | - | 0.23 .1-.4 | 0.21 .1-.4 | 0.20 .0-.5 |
| KP H | | | | - | 0.81 .7-.9 | 0.70 .5-.9 |
| KP HP | | | | | - | 0.70 .5-.9 |

| (c) | EC H | EC HP | EC F | KP H | KP HP | KP F |
|---|---|---|---|---|---|---|
| EC H | - | 0.86 .8-.9 | 0.70 .6-.8 | 0.37 .3-.5 | 0.43 .3-.5 | 0.30 .1-.5 |
| EC HP | | - | 0.72 .6-.8 | 0.29 .2-.4 | 0.35 .2-.5 | 0.25 .0-.5 |
| EC F | | | - | 0.18 .1-.3 | 0.25 .1-.4 | 0.14 .0-.5 |
| KP H | | | | - | 0.82 .7-.9 | 0.76 .6-.9 |
| KP HP | | | | | - | 0.68 .5-.8 |

| (d) | EC H | EC HP | EC F | KP H | KP HP | KP F |
|---|---|---|---|---|---|---|
| EC H | - | 0.84 .8-.9 | 0.56 .5-.6 | 0.68 .6-.7 | 0.71 .6-.8 | 0.60 .5-.7 |
| EC HP | | - | 0.66 .6-.7 | 0.65 .6-.7 | 0.68 .6-.7 | 0.62 .5-.7 |
| EC F | | | - | 0.42 .3-.5 | 0.46 .4-.6 | 0.47 .3-.6 |
| KP H | | | | - | 0.86 .8-.9 | 0.67 .5-.8 |
| KP HP | | | | | - | 0.64 .5-.8 |

**Legend**

Species
- EC — ESBL-producing *E. coli*
- KP — ESBL-producing *K. pneumoniae*

Reservoir
- H — human
- HP — human-polluted environment
- F — food

Acquired resistance gene class
- (a) beta-lactamase
- (b) ESBL
- (c) fluoroquinolone
- (d) aminoglycoside

PSI: 0, 0.1, 0.2, 0.3, 0.4, 0.5, 0.6, 0.7, 0.8, 0.9, 1

**Fig 2. Czekanowski's proportional similarity index (PSI) (95% confidence interval) of detected antimicrobial resistance gene (ARG) types.** Panel (a) beta-lactamases conferring resistance to penicillins only (small and broad spectrum), panel **(b)** ESBL, panel (c) fluoroquinolones, and panel (d) aminoglycosides. The PSI is calculated with the following formula: $1 - 0.5 \sum k \left| p(reservoir[n])k - q(reservoir[nx])k \right|$ where $p$ corresponded to the relative frequency of gene type $k$ (e.g. $bla_{CTX-M-15}$) in reservoir $n$ (e.g., ESBL-*E. coli* from humans), and $q$ corresponded to the relative frequency of the same gene type in reservoir $nx$ (e.g., ESBL-*K. pneumoniae* from humans). The numerator of the relative frequency was the count of each ARG type. The denominator of the relative frequency was the total number of genes of the corresponding ARG class. The PSI is a proportion, with 0 interpreted as no overlap, and 1 as perfect overlap in ARG type distributions between two reservoirs. 95% confidence intervals were calculated with 5,000 bootstrap iterations. PSI analysis for fosfomycin, trimethoprim, and sulfonamides ARGs are described in the supplement (S1 Fig 4 in S1 Appendix).

prospectively and longitudinally sampled from humans, the hp-environment, and food in five European metropolitan areas. Due to the sampling strategy, we were able to give unique insight in the co-occurrence of these species. Furthermore, the availability of short read sequences of all isolates enabled us to compare the assessed reservoirs on several genetic levels.

Several limitations must be acknowledged. Firstly, horizontal gene transfer of ARGs could not be assessed due to limitations of short-read sequencing technology in reconstructing mobile genetic elements such as plasmids [26]. Ongoing MODERN studies using previously completed nanopore sequencing aim to address plasmid characterization, overcoming any issues with repetitive sequence elements. Secondly, sample size and sampling strategies hampered comparisons between species and reservoirs. Fewer ESBL-Kp isolates were collected than ESBL-Ec, largely due to a lower prevalence, especially in food. Additionally, while LTCF food items came directly from participating LCTFs, household food items were sampled from nearby supermarkets. A higher similarity between human carriage and food may have been observed if the food items came directly from participating households. As a result, this study could not reliably determine the exact relation between contamination of specific food items and ESBL carriage in participating individuals. Thirdly, two different cgMLST thresholds were used: one to select unique intra-individual isolates, and one to determine clonal transmission

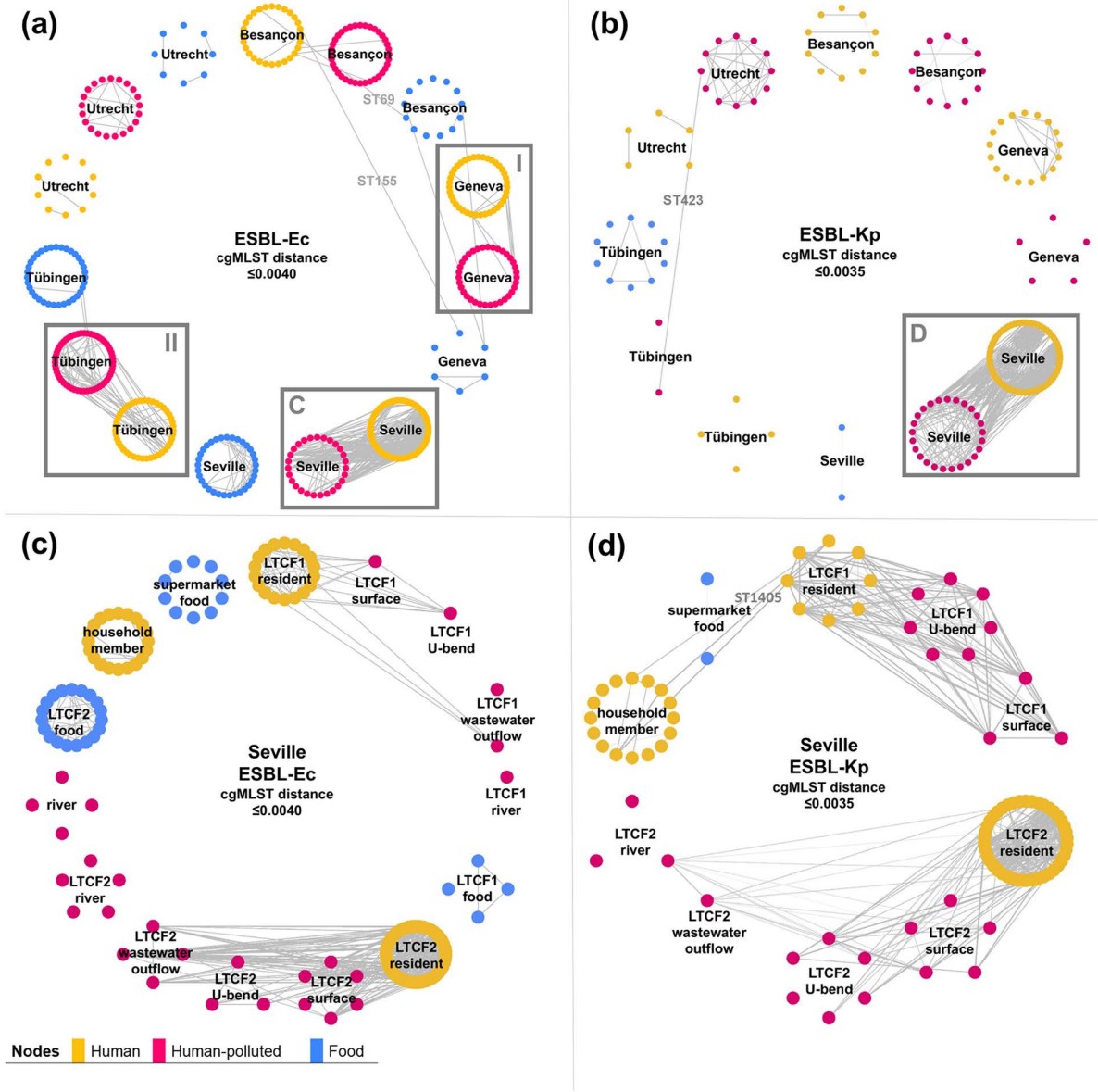

**Fig 3. Visualization of clonally related isolate pairs.** Nodes represent isolates and are grouped based on epidemiological setting, lines represent genetically similar isolate pairs. Panel **(a)** ESBL-producing *E. coli* (ESBL-Ec), panel **(b)** ESBL-producing *K. pneumoniae* (including seven *K. variicola isolates*) (ESBL-Kp), panel **(c)** ESBL-Ec Seville, panel **(d)** ESBL-Kp Seville. Grey boxes I and II are displayed enlarged in S1 Fig 5 in S1 Appendix, I: Panel A, II panel **B.**

between humans and non-human groups. Using a single threshold would have either overestimated the number of unique intra-individual isolates, or the clonal relationships between humans and non-human groups. The optimal threshold for the sampled groups likely varies across settings and remains unknown. Finally, the generalizability of the results to community dwellers is likely limited, due to the included health-care associated study population, where a higher ESBL prevalence, and thus clonal and horizontal transmission, may have led to overestimation of ARG similarity between ESBL-Ec and ESBL-Kp in humans.

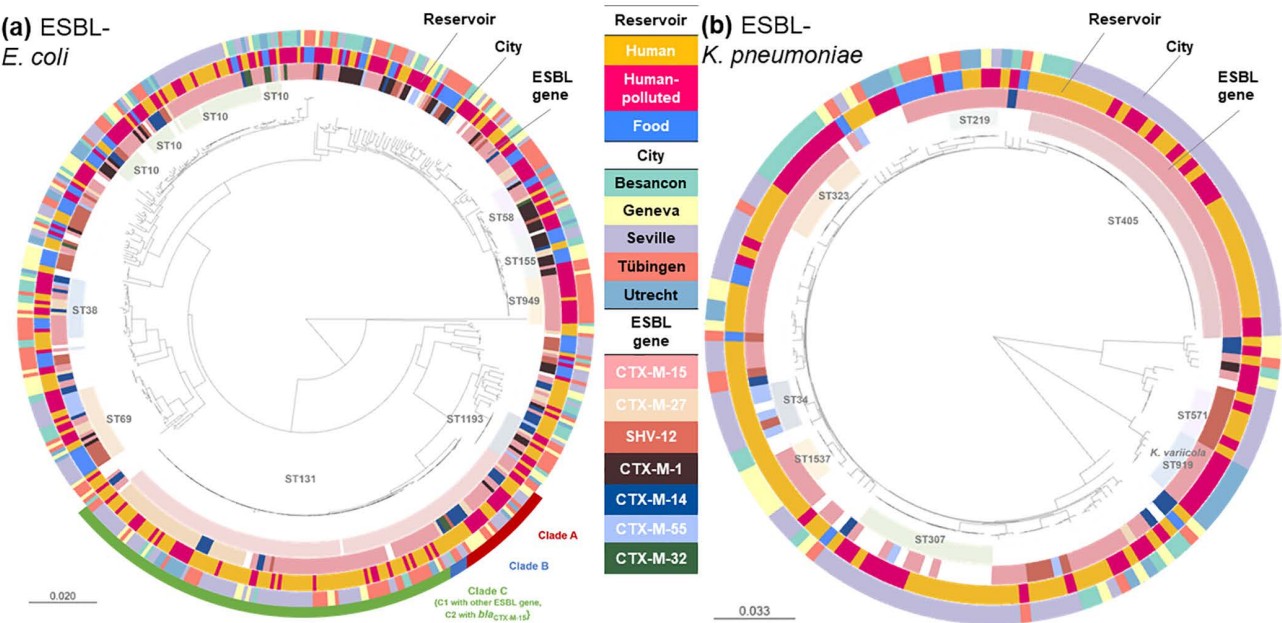

**Fig 4. Maximum likelihood core genome phylogenies.** Panel **(a)** ESBL-producing *E. coli* (ESBL-Ec) (*n* = 475 isolates), panel **(b)** ESBL-producing *K. pneumoniae* (ESBL-Kp) (*n* = 171 isolates, including seven *K. variicola* isolates). Online access interactive trees: ESBL-Ec: https://microreact.org/project/wSXSAfjgAH1Xr9af46Hbcb-modern-project-esbl-e-coli-n475.

Our data suggest the importance of public health interventions on prevention of human-to-human clonal and horizontal transmission through (i) implementation of adequate infection prevention in healthcare centers like LTCFs, and (ii) development of hygienic advice for households of colonized patients discharged from hospitals. Future research should elucidate to which extent, and under which circumstances, members of the *Enterobacterales* family share ARGs through HGT, and how this contributes to endemicity of ESBL-Ec and ESBL-Kp.

In conclusion, ESBL-Ec and ESBL-Kp regularly co-occurred in human populations with a close relationship with healthcare throughout Europe. Considerable overlap in ARG content was observed between ESBL-Ec and ESBL-Kp. Furthermore, clonal links were frequently observed between humans and the human-polluted environment, and at a lower frequency between humans and food. These findings are consistent with human-to-human transmission as an important driver of the dissemination of ARGs in humans.

## Supporting information

**S1 Appendix. Supplementary appendix.**
(DOCX)

**S1 Table. Strainfile.** Contains data of all included strains in this study.
(XLSX)

## Acknowledgments

We would like to thank the staff and residents from the LTCFs of Bellevaux in Besançon – France; Ferrusola and El Recreo in Seville – Spain; Luise-Wetzel-Stift and Samariterstift in Tübingen, Germany; and Altenahove in Almkerk – The Netherlands for their involvement in the study. We would like to thank Mercedes Delgado from Seville – Spain; Caroline

Brossier from Geneva – Switzerland, for assistance in collecting and processing the samples. We would like to thank Judith Vlooswijk, and Heike Schmitt from Utrecht – the Netherlands; Marion Broussier, Jeanne Celotto, and Alexandre Meunier from Besançon – France; John Poté, Gesuele Renzi, Abdessalam Cherkaoui, and Siva Lingam from Geneva, Switzerland; Michael Eib from Tübingen – Germany for their help on microbiological analyses of the samples, Elisabeth Stoll and Steffen Ganß from Tübingen – Germany for collecting the samples. We thank the Institut Pasteur teams for the curation and maintenance of BIGSdb-Pasteur databases at http://bigsdb.pasteur.fr/. We thank the Enterobase team for the curation and maintenance of Enterobase *Escherichia coli* MLST database at https://enterobase.warwick.ac.uk. We thank the academic editors and reviewers, for providing commentary and constructive feedback.

## Author contributions

**Conceptualization:** Maria Eugenia Riccio, Ingo B. Autenrieth, Ben S. Cooper, Didier Hocquet, Evelina Tacconelli, Jesus Rodriguez-Baño, Stephan Harbarth, Ad C. Fluit, Silke Peter, Jan A.J.W. Kluytmans.

**Data curation:** Tess D. Verschuuren, Julia Guther, Maria Eugenia Riccio, Daniel Martak, Elena Salamanca, Siri Göpel, Nadine Conzelmann, Jelle Scharringa, Silke Peter, Jan A.J.W. Kluytmans.

**Formal analysis:** Tess D. Verschuuren, Julia Guther, Jelle Scharringa, Ad C. Fluit, Silke Peter, Jan A.J.W. Kluytmans.

**Funding acquisition:** Ingo B. Autenrieth, Ben S. Cooper, Didier Hocquet, Evelina Tacconelli, Jesus Rodriguez-Baño, Stephan Harbarth, Silke Peter, Jan A.J.W. Kluytmans.

**Investigation:** Tess D. Verschuuren, Julia Guther, Jelle Scharringa, Ad C. Fluit, Silke Peter, Jan A.J.W. Kluytmans.

**Methodology:** Tess D. Verschuuren, Julia Guther, Jelle Scharringa, Patrick Musicha, Ben S. Cooper, Ad C. Fluit, Silke Peter, Jan A.J.W. Kluytmans.

**Project administration:** Tess D. Verschuuren, Julia Guther, Maria Eugenia Riccio, Daniel Martak, Elena Salamanca, Siri Göpel, Nadine Conzelmann, Patrick Musicha, Ingo B. Autenrieth, Didier Hocquet, Silke Peter, Jan A.J.W. Kluytmans.

**Resources:** Ingo B. Autenrieth, Ben S. Cooper, Didier Hocquet, Evelina Tacconelli, Jesus Rodriguez-Baño, Stephan Harbarth, Ad C. Fluit, Silke Peter, Jan A.J.W. Kluytmans.

**Software:** Tess D. Verschuuren, Julia Guther, Silke Peter, Jan A.J.W. Kluytmans.

**Supervision:** Didier Hocquet, Evelina Tacconelli, Jesus Rodriguez-Baño, Stephan Harbarth, Ad C. Fluit, Silke Peter, Jan A.J.W. Kluytmans.

**Validation:** Tess D. Verschuuren, Julia Guther, Jelle Scharringa, Ad C. Fluit, Silke Peter, Jan A.J.W. Kluytmans.

**Visualization:** Tess D. Verschuuren, Julia Guther.

**Writing – original draft:** Tess D. Verschuuren, Julia Guther, Ad C. Fluit, Silke Peter, Jan A.J.W. Kluytmans.

**Writing – review & editing:** Tess D. Verschuuren, Julia Guther, Maria Eugenia Riccio, Daniel Martak, Elena Salamanca, Siri Göpel, Nadine Conzelmann, Jelle Scharringa, Patrick Musicha, Ingo B. Autenrieth, Ben S. Cooper, Didier Hocquet, Evelina Tacconelli, Jesus Rodriguez-Baño, Stephan Harbarth, Ad C. Fluit, Silke Peter, Jan A.J.W. Kluytmans.

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
