## [Decision Letter · Decision Letter 0]

21 Jul 2025

Dear Dr. Verschuuren,

We look forward to receiving your revised manuscript.

Kind regards,

Gabriel Trueba, PhD

Academic Editor

PLOS ONE

Journal Requirements:

[The MODERN studies (Understanding and modelling reservoirs, vehicles and transmission of ESBL-producing Enterobacteriaceae in the community and long term care facilities) were part of a Joint Programming Initiative on Antimicrobial Resistance collaborative  research project, under the 2016 Joint Call framework (Transnational Research Projects on the Transmission Dynamics of Antibacterial Resistance). It received funding from the following national research agencies: Instituto de Salud Carlos III (grant no. AC16/00076), Netherlands Organization for Health Research and Development (grant no. 681055, 547001004), Swiss National Science Foundation (grant no. 40AR40-173608), German Federal Ministry of Education and Research (grant no. 01KI1701), the French Agence Nationale de la Recherche (grant no. ANR-16-JPEC-0007-03), and UK Medical Research Council (grant no. MR/R004536/1). Elena Salamanca, and Jesús Rodríguez-Baño received support for research from by the Plan Nacional de I+D+i 2013-2016 and Instituto de Salud Carlos III, Subdirección General de Redes y Centros de Investigación Cooperativa, Ministerio de Ciencia, Innovación y Universidades, Spanish Network for Research in Infectious Diseases (REIPI RD16/0016/0001), co-financed by the European Development Regional Fund A way to achieve Europe, Operative Program Intelligence Growth 2014–2020.].

[The MODERN studies (Understanding and modelling reservoirs, vehicles and transmission of ESBL-producing Enterobacteriaceae in the community and long term care facilities) were part of a Joint Programming Initiative on Antimicrobial Resistance collaborative  research project, under the 2016 Joint Call framework (Transnational Research Projects on the Transmission Dynamics of Antibacterial Resistance). It received funding from the following national research agencies: Instituto de Salud Carlos III (grant no. AC16/00076), Netherlands Organization for Health Research and Development (grant no. 681055, 547001004), Swiss National Science Foundation (grant no. 40AR40-173608), German Federal Ministry of Education and Research (grant no. 01KI1701), the French Agence Nationale de la Recherche (grant no. ANR-16-JPEC-0007-03), and UK Medical Research Council (grant no. MR/R004536/1). Elena Salamanca, and Jesús Rodríguez-Baño received support for research from by the Plan Nacional de I+D+i 2013-2016 and Instituto de Salud Carlos III, Subdirección General de Redes y Centros de Investigación Cooperativa, Ministerio de Ciencia, Innovación y Universidades, Spanish Network for Research in Infectious Diseases (REIPI RD16/0016/0001), co-financed by the European Development Regional Fund A way to achieve Europe, Operative Program Intelligence Growth 2014–2020.].

5. Thank you for stating the following in your manuscript:

[The MODERN studies (Understanding and modelling reservoirs, vehicles and transmission of ESBL-producing Enterobacteriaceae in the community and long term care facilities) were part of a Joint Programming Initiative on Antimicrobial Resistance collaborative  research project, under the 2016 Joint Call framework (Transnational Research Projects on the Transmission Dynamics of Antibacterial Resistance). It received funding from the following national research agencies: Instituto de Salud Carlos III (grant no. AC16/00076), Netherlands Organization for Health Research and Development (grant no. 681055, 547001004), Swiss National Science Foundation (grant no. 40AR40-173608), German Federal Ministry of Education and Research (grant no. 01KI1701), the French Agence Nationale de la Recherche (grant no. ANR-16-JPEC-0007-03), and UK Medical Research Council (grant no. MR/R004536/1). Elena Salamanca, and Jesús Rodríguez-Baño received support for research from by the Plan Nacional de I+D+i 2013-2016 and Instituto de Salud Carlos III, Subdirección General de Redes y Centros de Investigación Cooperativa, Ministerio de Ciencia, Innovación y Universidades, Spanish Network for Research in Infectious Diseases (REIPI RD16/0016/0001), co-financed by the European Development Regional Fund A way to achieve Europe, Operative Program Intelligence Growth 2014–2020.]

[The MODERN studies (Understanding and modelling reservoirs, vehicles and transmission of ESBL-producing Enterobacteriaceae in the community and long term care facilities) were part of a Joint Programming Initiative on Antimicrobial Resistance collaborative  research project, under the 2016 Joint Call framework (Transnational Research Projects on the Transmission Dynamics of Antibacterial Resistance). It received funding from the following national research agencies: Instituto de Salud Carlos III (grant no. AC16/00076), Netherlands Organization for Health Research and Development (grant no. 681055, 547001004), Swiss National Science Foundation (grant no. 40AR40-173608), German Federal Ministry of Education and Research (grant no. 01KI1701), the French Agence Nationale de la Recherche (grant no. ANR-16-JPEC-0007-03), and UK Medical Research Council (grant no. MR/R004536/1). Elena Salamanca, and Jesús Rodríguez-Baño received support for research from by the Plan Nacional de I+D+i 2013-2016 and Instituto de Salud Carlos III, Subdirección General de Redes y Centros de Investigación Cooperativa, Ministerio de Ciencia, Innovación y Universidades, Spanish Network for Research in Infectious Diseases (REIPI RD16/0016/0001), co-financed by the European Development Regional Fund A way to achieve Europe, Operative Program Intelligence Growth 2014–2020.].

6. Please amend the manuscript submission data (via Edit Submission) to include authors TD Verschuuren, J Guther, ME Riccio, D Martak, E Salamanca, S Göpel, N Conzelmann, J Scharringa, P Musicha, IB Autenrieth, BS Cooper, D Hocquet, E Tacconelli, J Rodriguez-Baño, S Harbarth, AC Fluit, S Peter, and JAJW Kluytmans.

7. Please amend your authorship list in your manuscript file to include authors Tess Verschuuren, Julia Guther, Maria Eugenia Riccio, Daniel Martak, Elena Salamanca, Siri Göpel, Nadine Conzelmann, Jelle Scharringa, Patrick Musicha, Ingo B. Autenrieth, Ben S. Cooper, Didier Hocquet, Evelina Tacconelli, Jesus Rodriguez-Baño, Stephan Harbarth, Ad C. Fluit, Silke Peter, and Jan A.J.W. Kluytmans.

Reviewers' comments:

Reviewer's Responses to Questions

**Comments to the Author**

1. Is the manuscript technically sound, and do the data support the conclusions?

Reviewer #1: Yes

Reviewer #2: Yes

2. Has the statistical analysis been performed appropriately and rigorously?

Reviewer #1: Yes

Reviewer #2: Yes

3. Have the authors made all data underlying the findings in their manuscript fully available?

Reviewer #1: Yes

Reviewer #2: Yes

4. Is the manuscript presented in an intelligible fashion and written in standard English?

Reviewer #1: Yes

Reviewer #2: Yes

Reviewer #1: The authors present a study on the epidemiology of extended-spectrum beta-lactamase (ESBL)-producing E. coli and K. pneumoniae. The work is based on isolates collected from five European metropolitan areas, which were genome-sequenced and analyzed using in silico approaches. Analyses of antimicrobial resistance gene (ARG) content, core genome multilocus sequence typing (cgMLST), and core genome comparisons allowed the authors to explore clonal relationships between isolates. The topic is of public health relevance, and the manuscript is generally well-written and easy to follow.

Minor comments:

The possible role of the hospital or LTCF environment as a reservoir or amplifier of multidrug-resistant bacteria is not discussed. This would be particularly relevant given the healthcare-associated nature of the study population.

The limitations section occupies more than half of the discussion and could be slightly condensed to improve balance with the rest of the text.

While the authors mention the need for long-read sequencing, they could elaborate more specifically on the types of questions that such technology would enable (e.g., plasmid structure, transmission dynamics of mobile genetic elements).

Would the authors consider routine genomic surveillance in hospitals or LTCFs useful? This could be suggested as a potential recommendation for future practice.

Reviewer #2: The study addresses a very important and pertinent issue. Epidemiological, laboratory and statistical methodology was great and innovative. As far as results are concerned, there was a bit if discrepancy between the text on line 158 and the table where text says Table 1 shows mrc-1 gene in an ESBL-EC strain while text indicates it is in an ESBL- Kp!!! Otherwise the results are well represented and easy to understand. The discussion analyses the results notes the limitations. Overall study is well designed, done and excellently written.

**Do you want your identity to be public for this peer review?** For information about this choice, including consent withdrawal, please see our Privacy Policy

Reviewer #1: No

Reviewer #2: **Yes: ** Henry Mawerere Kajumbula

---

## [Author Response · Author response to Decision Letter 1]

22 Oct 2025

Journal Requirements:

We adjusted the manuscript based on PLOS ONE’s style requirements. Including font, lay-out of headers, from British English to American English, and reference style.

[The MODERN studies (Understanding and modelling reservoirs, vehicles and transmission of ESBL-producing Enterobacteriaceae in the community and long term care facilities) were part of a Joint Programming Initiative on Antimicrobial Resistance collaborative research project, under the 2016 Joint Call framework (Transnational Research Projects on the Transmission Dynamics of Antibacterial Resistance). It received funding from the following national research agencies: Instituto de Salud Carlos III (grant no. AC16/00076), Netherlands Organization for Health Research and Development (grant no. 681055, 547001004), Swiss National Science Foundation (grant no. 40AR40-173608), German Federal Ministry of Education and Research (grant no. 01KI1701), the French Agence Nationale de la Recherche (grant no. ANR-16-JPEC-0007-03), and UK Medical Research Council (grant no. MR/R004536/1). Elena Salamanca, and Jesús Rodríguez-Baño received support for research from by the Plan Nacional de I+D+i 2013-2016 and Instituto de Salud Carlos III, Subdirección General de Redes y Centros de Investigación Cooperativa, Ministerio de Ciencia, Innovación y Universidades, Spanish Network for Research in Infectious Diseases (REIPI RD16/0016/0001), co-financed by the European Development Regional Fund A way to achieve Europe, Operative Program Intelligence Growth 2014–2020.].

We have amended the funding statement, and included ‘there was no additional external funding received for this study, see below.

Funding statement

The MODERN studies (Understanding and modelling reservoirs, vehicles and transmission of ESBL-producing Enterobacteriaceae in the community and long term care facilities) were part of a Joint Programming Initiative on Antimicrobial Resistance collaborative research project, under the 2016 Joint Call framework (Transnational Research Projects on the Transmission Dynamics of Antibacterial Resistance). It received funding from the following national research agencies: Instituto de Salud Carlos III (grant no. AC16/00076), Netherlands Organization for Health Research and Development (grant no. 681055, 547001004), Swiss National Science Foundation (grant no. 40AR40-173608), German Federal Ministry of Education and Research (grant no. 01KI1701), the French Agence Nationale de la Recherche (grant no. ANR-16-JPEC-0007-03), and UK Medical Research Council (grant no. MR/R004536/1). Elena Salamanca, and Jesús Rodríguez-Baño received support for research from by the Plan Nacional de I+D+i 2013-2016 and Instituto de Salud Carlos III, Subdirección General de Redes y Centros de Investigación Cooperativa, Ministerio de Ciencia, Innovación y Universidades, Spanish Network for Research in Infectious Diseases (REIPI RD16/0016/0001), co-financed by the European Development Regional Fund A way to achieve Europe, Operative Program Intelligence Growth 2014–2020. There was no additional external funding received for this study.

[The MODERN studies (Understanding and modelling reservoirs, vehicles and transmission of ESBL-producing Enterobacteriaceae in the community and long term care facilities) were part of a Joint Programming Initiative on Antimicrobial Resistance collaborative research project, under the 2016 Joint Call framework (Transnational Research Projects on the Transmission Dynamics of Antibacterial Resistance). It received funding from the following national research agencies: Instituto de Salud Carlos III (grant no. AC16/00076), Netherlands Organization for Health Research and Development (grant no. 681055, 547001004), Swiss National Science Foundation (grant no. 40AR40-173608), German Federal Ministry of Education and Research (grant no. 01KI1701), the French Agence Nationale de la Recherche (grant no. ANR-16-JPEC-0007-03), and UK Medical Research Council (grant no. MR/R004536/1). Elena Salamanca, and Jesús Rodríguez-Baño received support for research from by the Plan Nacional de I+D+i 2013-2016 and Instituto de Salud Carlos III, Subdirección General de Redes y Centros de Investigación Cooperativa, Ministerio de Ciencia, Innovación y Universidades, Spanish Network for Research in Infectious Diseases (REIPI RD16/0016/0001), co-financed by the European Development Regional Fund A way to achieve Europe, Operative Program Intelligence Growth 2014–2020.].

We have amended ‘Role of Funder statement’

Role of Funder statement

5. Thank you for stating the following in your manuscript:

[The MODERN studies (Understanding and modelling reservoirs, vehicles and transmission of ESBL-producing Enterobacteriaceae in the community and long term care facilities) were part of a Joint Programming Initiative on Antimicrobial Resistance collaborative research project, under the 2016 Joint Call framework (Transnational Research Projects on the Transmission Dynamics of Antibacterial Resistance). It received funding from the following national research agencies: Instituto de Salud Carlos III (grant no. AC16/00076), Netherlands Organization for Health Research and Development (grant no. 681055, 547001004), Swiss National Science Foundation (grant no. 40AR40-173608), German Federal Ministry of Education and Research (grant no. 01KI1701), the French Agence Nationale de la Recherche (grant no. ANR-16-JPEC-0007-03), and UK Medical Research Council (grant no. MR/R004536/1). Elena Salamanca, and Jesús Rodríguez-Baño received support for research from by the Plan Nacional de I+D+i 2013-2016 and Instituto de Salud Carlos III, Subdirección General de Redes y Centros de Investigación Cooperativa, Ministerio de Ciencia, Innovación y Universidades, Spanish Network for Research in Infectious Diseases (REIPI RD16/0016/0001), co-financed by the European Development Regional Fund A way to achieve Europe, Operative Program Intelligence Growth 2014–2020.]

[The MODERN studies (Understanding and modelling reservoirs, vehicles and transmission of ESBL-producing Enterobacteriaceae in the community and long term care facilities) were part of a Joint Programming Initiative on Antimicrobial Resistance collaborative research project, under the 2016 Joint Call framework (Transnational Research Projects on the Transmission Dynamics of Antibacterial Resistance). It received funding from the following national research agencies: Instituto de Salud Carlos III (grant no. AC16/00076), Netherlands Organization for Health Research and Development (grant no. 681055, 547001004), Swiss National Science Foundation (grant no. 40AR40-173608), German Federal Ministry of Education and Research (grant no. 01KI1701), the French Agence Nationale de la Recherche (grant no. ANR-16-JPEC-0007-03), and UK Medical Research Council (grant no. MR/R004536/1). Elena Salamanca, and Jesús Rodríguez-Baño received support for research from by the Plan Nacional de I+D+i 2013-2016 and Instituto de Salud Carlos III, Subdirección General de Redes y Centros de Investigación Cooperativa, Ministerio de Ciencia, Innovación y Universidades, Spanish Network for Research in Infectious Diseases (REIPI RD16/0016/0001), co-financed by the European Development Regional Fund A way to achieve Europe, Operative Program Intelligence Growth 2014–2020.].

We removed the funding statement from the manuscript.

6. Please amend the manuscript submission data (via Edit Submission) to include authors TD Verschuuren, J Guther, ME Riccio, D Martak, E Salamanca, S Göpel, N Conzelmann, J Scharringa, P Musicha, IB Autenrieth, BS Cooper, D Hocquet, E Tacconelli, J Rodriguez-Baño, S Harbarth, AC Fluit, S Peter, and JAJW Kluytmans.

It’s not clear what needs to be amended here, the author list is correct. Should first names be replaced to first letter only? The author list form in the submission data in the editorial manager page seems to request full first names.

7. Please amend your authorship list in your manuscript file to include authors Tess Verschuuren, Julia Guther, Maria Eugenia Riccio, Daniel Martak, Elena Salamanca, Siri Göpel, Nadine Conzelmann, Jelle Scharringa, Patrick Musicha, Ingo B. Autenrieth, Ben S. Cooper, Didier Hocquet, Evelina Tacconelli, Jesus Rodriguez-Baño, Stephan Harbarth, Ad C. Fluit, Silke Peter, and Jan A.J.W. Kluytmans.

We amended authorship list to

Tess D. Verschuuren1,2, Julia Guther3, Maria Eugenia Riccio4, Daniel Martak5,6, Elena Salamanca7, Siri Göpel8, Nadine Conzelmann8, Jelle Scharringa9, Patrick Musicha10, Ingo B. Autenrieth11, Ben S. Cooper10, Didier Hocquet5,6, Evelina Tacconelli8,12, Jesús Rodríguez-Baño7, Stephan Harbarth4, Ad C. Fluit9, Silke Peter3, Jan A. J. W. Kluytmans2,9

A supporting information section was added to the manuscript

Not applicable

Done

Reviewers' comments:

Reviewer #1: The authors present a study on the epidemiology of extended-spectrum beta-lactamase (ESBL)-producing E. coli and K. pneumoniae. The work is based on isolates collected from five European metropolitan areas, which were genome-sequenced and analyzed using in silico approaches. Analyses of antimicrobial resistance gene (ARG) content, core genome multilocus sequence typing (cgMLST), and core genome comparisons allowed the authors to explore clonal relationships between isolates. The topic is of public health relevance, and the manuscript is generally well-written and easy to follow.

Reviewer 1, thank you for taking the time to review our manuscript.

Minor comments:

The possible role of the hospital or LTCF environment as a reservoir or amplifier of multidrug-resistant bacteria is not discussed. This would be particularly relevant given the healthcare-associated nature of the study population.

We have added the following sentence to the discussion (L265) ‘Hospital and LTCF environments are well-established as significant reservoirs and amplifiers of multi-drug resistant bacteria, contributing to the persistence and spread of these organisms within healthcare settings and the broader community [16, 24, 25].’

The limitations section occupies more than half of the discussion and could be slightly condensed to improve balance with the rest of the text.

Agreed, we have shortened the limitations section to about 2/3rds of the original length.

While the authors mention the need for long-read sequencing, they could elaborate more specifically on the types of questions that such technology would enable (e.g., plasmid structure, transmission dynamics of mobile genetic elements).

Thank you for this suggestion. We find that this point is the main limitation of the study, and future results from nanopore sequencing analyses are of interest to the scientific community.

We have added a few details on challenges with assembly due to repetitive sequences (L272), but refrained from adding more information on long-read sequencing for the sake of brevity. ‘Several limitations must be acknowledged. Firstly, horizontal gene transfer of ARGs could not be assessed due to limitations of short-read sequencing technology in reconstructing mobile genetic elements such as plasmids [26]. Ongoing MODERN studies using previously completed nanopore sequencing aim to address plasmid characterization, overcoming any issues with repetitive sequence elements.’

Would the authors consider routine genomic surveillance in hospitals or LTCFs useful? This could be suggested as a potential recommendation for future practice.

As the consideration of genomic surveillance as a routine procedure would ensuite a more detailed discussion about the appropriateness of the effort and the benefits, and should furthermore incorporate discussion about AMR in mobile genetic elements, we refrain from elaborating such recommendations within this manuscript.

Reviewer #2: The study addresses a very important and pertinent issue. Epidemiological, laboratory and statistical methodology was great and innovative. As far as results are concerned, there was a bit if discrepancy between the text on line 158 and the table where text says Table 1 shows mrc-1 gene in an ESBL-EC strain while text indicates it is in an ESBL- Kp!!! Otherwise the results are well represented and easy to understand. The discussion analyses the results notes the limitations. Overall study is well designed, done and excellently written.

Reviewer 2, thank you for taking the time to read and comment on our MS, many thanks for pointing this discrepancy out. The mrc-1 gene indeed belonged to an ESBL-Kp from food collected in Tübingen (isolate_625 in S2_Table). Table 2 was incorrect and has been adjusted accordingly.

---

## [Decision Letter · Decision Letter 1]

6 Nov 2025

Extended-spectrum beta-lactamase-producing Escherichia coli and Klebsiella pneumoniae from human carriage, the human-polluted environment, and food: molecular epidemiology of two prospective cohorts in five European metropolitan areas

PONE-D-25-20088R1

Dear Dr. Verschuuren,

We’re pleased to inform you that your manuscript has been judged scientifically suitable for publication and will be formally accepted for publication once it meets all outstanding technical requirements.

Kind regards,

Gabriel Trueba, PhD

Academic Editor

PLOS ONE

Additional Editor Comments (optional):

Reviewers' comments:

Reviewer's Responses to Questions

**Comments to the Author**

Reviewer #1: All comments have been addressed

2. Is the manuscript technically sound, and do the data support the conclusions?

Reviewer #1: Yes

3. Has the statistical analysis been performed appropriately and rigorously?

Reviewer #1: Yes

4. Have the authors made all data underlying the findings in their manuscript fully available?

Reviewer #1: Yes

5. Is the manuscript presented in an intelligible fashion and written in standard English?

Reviewer #1: Yes

Reviewer #1: (No Response)

**Do you want your identity to be public for this peer review?** For information about this choice, including consent withdrawal, please see our Privacy Policy

Reviewer #1: No

---

## [Editor Report · Acceptance letter]

PONE-D-25-20088R1

PLOS One

Dear Dr. Verschuuren,

I'm pleased to inform you that your manuscript has been deemed suitable for publication in PLOS One. Congratulations! Your manuscript is now being handed over to our production team.

Kind regards,

on behalf of

Dr. Gabriel Trueba

Academic Editor

PLOS One